# LIT: Block-wise Intermediate Representation Training for Model Compression

## Abstract

Knowledge distillation (KD) is a popular method for reducing the computational overhead of deep network inference, in which the output of a teacher model is used to train a smaller, faster student model. Hint training (i.e., FitNets) extends KD by regressing a student model's intermediate representation to a teacher model's intermediate representation. In this work, we introduce bLock-wise Intermediate representation Training (LIT), a novel model compression technique that extends the use of intermediate representations in deep network compression, outperforming KD and hint training. LIT has two key ideas: 1) LIT trains a student of the same width (but shallower depth) as the teacher by directly comparing the intermediate representations, and 2) LIT uses the intermediate representation from the previous block in the teacher model as an input to the current student block during training, avoiding unstable intermediate representations in the student network. We show that LIT provides substantial reductions in network depth *without loss in accuracy* — for example, LIT can compress a ResNeXt-110 to a ResNeXt-20 (5.5×) on CIFAR10 and a VDCNN-29 to a VDCNN-9 (3.2×) on Amazon Reviews without loss in accuracy, outperforming KD and hint training in network size for a given accuracy. We also show that applying LIT to identical student/teacher architectures increases the accuracy of the student model above the teacher model, outperforming the recently-proposed Born Again Networks procedure on ResNet, ResNeXt, and VDCNN. Finally, we show that LIT can effectively compress GAN generators.

## 1 Introduction

Modern deep networks have achieved increased accuracy by continuing to introduce more layers (Ioffe & Szegedy, 2015; He et al., 2016) at the cost of higher computational overhead. In response, researchers have proposed many techniques to reduce this computational overhead at inference time, which broadly fall under two categories. First, in deep compression (Han et al., 2015a; Zhu et al., 2016; Li et al., 2016; Hubara et al., 2017), parts of a model are removed or quantized to reduce the number of weights and/or the computational footprint.[1] However, deep compression techniques typically require new hardware (Han et al., 2016) to take advantage of the resulting model sparsity. Second, in student/teacher methods—introduced in knowledge distillation (KD) (Hinton et al., 2014) and further extended (Romero et al., 2015; Kim & Rush, 2016; Furlanello et al., 2018)—a smaller student model learns from a large teacher model through distillation loss, wherein the student model attempts to match the logits of the teacher model. As there are no constraints on the teacher and student models, KD can produce hardware-friendly models: the student can be a standard model architecture (e.g., ResNet), optimized for a given hardware substrate.

Hint training (i.e., FitNets (Romero et al., 2015)) extends KD by using a teacher's intermediate representation (IR, i.e., the output from a hidden layer) to guide the training of the student model. The authors show that hint training with a single IR outperforms KD in compressing teacher networks (e.g., maxout networks (Goodfellow et al., 2013)) to thinner and deeper student networks.

We ask the natural question: does hint training compress more modern, highly-structured, very deep networks—such as ResNet (He et al., 2016), ResNeXt (Xie et al., 2017), VDCNN (Conneau et al., 2016), and StarGAN (Choi et al., 2017)? We find that standard hint training (i.e., with a single hint) and training with multiple hints is not effective for modern deep networks (Section 4.2). We hypothesize that, for

---

[1] In this work, we refer to this class of methods as "deep compression," and methods to reduce model size more generally as "model compression."

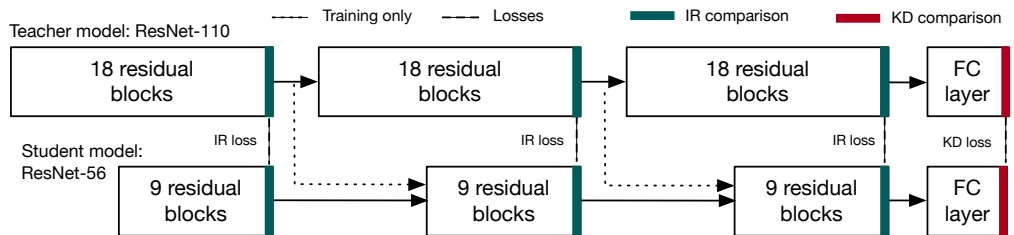

Figure 1: A schematic of LIT. In LIT, the teacher model's blocks are used as input to the student model's blocks during training, except for the first block. Specifically, denoting the blocks $S_1,...,S_4$ for the student and $T_1,...,T_4$ for the student, $S_2(T_1)$ is compared against $T_2$ in training and similarly for deeper parts of the network. $S_1$ and $T_1$ are directly compared. LIT additionally compares $S$ and $T$ through the KD loss. The teacher model is not updated in training.

modern deep networks, hint training causes unstable IRs: the deepest network considered in Romero et al. (2015) was only 17 layers, achieving 91.61% on CIFAR10; in contrast, a modern 110-layer ResNet achieves 93.68% on CIFAR10.

In this work, we extend hint training's ability to transfer intermediate knowledge from teacher to student to reduce the depth of modern, highly-structured architectures (e.g., compressing a standard ResNeXt-110 to a standard ResNeXt-20 with no loss in accuracy). We do this via a novel method called b**L**ock-wise **I**ntermediate representation **T**raining (LIT), a student/teacher compression technique that outperforms training student networks from scratch, hint training, and KD. LIT targets highly structured, modern networks that consist of repetitive blocks (i.e., groups of layers) that can be scaled up/down for accuracy/speed trade-offs; for example, ResNets have standard configurations from 20 to hundreds of layers. LIT leverages two key ideas to reduce unstable IRs in deep networks. First, LIT directly trains student networks of the *same width* as the teacher model (as opposed to using a single, thinner hint as in hint training). Second, LIT avoids unstable student IRs deep in the network by using the IR from the *previous* block in the teacher model as input to the current student block during training; each student block is effectively trained in isolation to match the corresponding (deeper) block in the teacher. We show that LIT's block-wise training improves accuracy, allows for copying parts of the teacher model directly to the student model, and permits selective compression of networks (e.g., compressing one out of three blocks in a network and copying the rest). For example, consider compressing a ResNet-56 from a ResNet-110 (Figure 1), each of which have four stages. The IR loss is applied to the output of each block, and the teacher model's IRs are used as input to the student blocks.

We show that LIT outperforms standard KD on a range of models (ResNet, ResNeXt, VDCNN, StarGAN) and datasets (CIFAR10, CIFAR100, Amazon Reviews, CelebA): empirically, LIT can reduce model sizes from $1.7\times$ to $5.5\times$ with no loss in accuracy. Recent work on Born Again networks (Furlanello et al., 2018) uses standard KD to train identical student and teacher models to higher accuracies (i.e., no compression). We show that the benefits of this procedure also apply to LIT student/teacher training, and LIT enables up to 0.64% higher accuracy than KD-based Born Again networks on the networks we consider.

## 2 RELATED WORK

**Knowledge distillation.** Hinton et al. (2014); Bucilu et al. (2006) introduced knowledge distillation (KD) in which a teacher ensemble or models outputs are used to train a smaller student model, which inspired a variety of related methods, e.g., for cross-modal distillation or faster training (Gupta et al., 2016; Chen et al., 2015; Frosst & Hinton, 2017; Romero et al., 2015; Furlanello et al., 2018). FitNets extends KD by regressing a student model's IR to a teacher model's IR, as the student models they consider are thinner and deeper. Wang et al. (2018) extends FitNets by training networks iteratively using hints. In contrast, LIT uses the teacher IRs as input to the student model in training and directly penalizes deviations of the student model's IRs from teacher model's IRs, which helps guide training for higher accuracy and improved inference performance. In Born Again networks (Furlanello et al., 2018), the same network architecture is used as both the teacher and student in standard KD, resulting in higher accuracy. We show that LIT outperforms the Born Again procedure on ResNet, ResNeXt, and VDCNN.

**Deep compression.** In deep compression, parts of a network (weights, groups of weights, kernels, or filters (Mao et al., 2017)) are removed for efficient inference (Han et al., 2015a), and the weights of the

network are quantized, hashed, or compressed (Hubara et al., 2016; Rastegari et al., 2016; Zhu et al., 2016; Hubara et al., 2017). These methods largely do not take advantage of a teacher model and typically require new hardware for efficiency gains (Han et al., 2016). Methods that prune filters (Li et al., 2016) can result in speedups on existing hardware, but largely degrade accuracy. We show that LIT models can be pruned, and thus these methods are complementary to LIT.

Additionally, deep compression does not perform as well on modern networks: Han et al. (2015a) compressed VGG by ~10×, but ResNet-110 can only be compressed ~1.6× (Li et al., 2016), compared to LIT's 4× compression.

**Network architectures for fast inference.** Researchers have proposed network architectures (e.g., MobileNet (Howard et al., 2017)) and new operations for fast inference (e.g., ShuffleNet (Zhang et al., 2017)) on specific hardware. However, these architectures and operations are largely designed for power/resource-constrained mobile devices and sacrifice accuracy for low power. We focus on highly accurate, very deep networks in this work.

## 3 METHODS

LIT uses an augmented loss function and training procedure to distill a teacher model into a student model. In its training procedure, LIT both 1) penalizes deviations of the student model's IRs from the teacher model's IRs (IR loss) and 2) uses the KD loss (for the entire student network). As LIT directly penalizes deviations in IRs, LIT requires that the teacher model and student model have outputs of the same size at some intermediate layer.

A key challenge in the LIT procedure is that the student network will not have meaningful IRs for a large part of the training (e.g., at the start of training when the weights are initialized randomly). To address this issue, LIT uses the teacher model's IRs as inputs to the student model (described below).

We describe the overall LIT procedure, describe the KD loss, describe how IRs are used in LIT, and discuss hyperparameter optimization.

**LIT.** In LIT, we combine the KD and IR loss. We show that combining the losses results in smaller models for a fixed accuracy in Section 4. Specifically, for teacher $T$ and student $S$ the full LIT loss is:

$$\beta \cdot \mathcal{L}_{\text{KD},\alpha}(T,S) + (1-\beta) \cdot \mathcal{L}_{\text{I}}(T,S) \tag{1}$$

with $\alpha, \beta \in [0,1]$ ($\alpha$ is described below, $\beta$ is an interpolation parameter). In some cases, we use $\beta = 0$, i.e., we only use the IR loss (e.g., for GANs).

As the IRs have matching dimensions, LIT also allows parts of the teacher model to be copied directly into the student model. For example, for ResNets, we copy the teacher's first convolution (before the skip connections) and fully connected layer to the student model. LIT can also be used to compress specific parts of a model, as we do with StarGAN's generator (Choi et al., 2017).

Finally, after we train the student model with combined loss, we fine-tune the student model with the original loss (KD loss for classification, generator loss for GANs).

**Knowledge distillation loss.** In KD, a (typically larger) teacher model or ensemble is used to train a (typically smaller) student model. Specifically, the KL-divergence between the probabilities of the student and teacher model is minimized, in addition to the standard cross-entropy loss.

Formally, denote (for the teacher model) $q_i^\tau = \frac{\exp(z_i/\tau)}{\sum_j z_j/\tau}$ where $z_i$ are the inputs to the softmax and $\tau$ is a hyperparmeter that "softens" the distribution. Denote $p_i^\tau$ to be the corresponding quantity for the student model.

Then, the full KD loss is:

$$\mathcal{L}_{\text{KD}}(p^\tau, q^\tau, y) = \alpha \cdot H(y, p^\tau) + (1-\alpha) \cdot H(p^\tau, q^\tau) \tag{2}$$

for $y$ to be the true labels, $H$ to be the cross-entropy loss, and $\alpha$ to be the interpolation parameter.

Hinton et al. (2014) sets $\alpha = 0.5$, but we show that the choice of $\alpha$ can affect performance (Section 4.4).

**Training via intermediate representations.** In LIT, we logically divide the student and teacher networks into $k$ sub-networks such that the input and output dimensions match for the corresponding sub-networks (an example is shown in Figure 1).

| Dataset | Task | Models |
|---|---|---|
| CIFAR10 | Image classification | ResNet, ResNeXt |
| CIFAR100 | Image classification | ResNet, ResNeXt |
| Amazon Reviews | Sentiment analysis (full, polarity) | VDCNN |
| CelebA | Image-to-image translation | StarGAN |

Table 1: List of datasets, tasks, and models for standard tasks that we compress with LIT.

Denote the full teacher network and student network to be $T$ and $S$ respectively. Denote the teacher sub-networks to be $T_i$ and the student sub-networks to be $S_i$ such that $T_i, S_i : \mathbb{R}^{n_{i-i}} \to \mathbb{R}^{n_i}$ and that the composition of the sub-networks is the full network, namely that $T_k(T_{k-1}(\cdots T_1(x))) = T(x)$. We will omit the argument when convenient.

Denote the loss on the IR loss $l$ (e.g., L2 loss). The full intermediate loss (given the set of splits) is:

$$\mathcal{L}_1(T,S) := l(S_1, T_1) + \sum_{i=2}^{k} l(S_i(T_{i-1}), T_i) \tag{3}$$

Concretely, consider a ResNet-110 as the teacher and a ResNet-56 as the student, each with three "stages", i.e., layers in the network with downsampling, and an L2 intermediate loss. Here, the first teacher ResNet "stage" is $T_1$, etc. and the L2 deviation from the feature maps, across all the downsampling feature maps, is the full intermediate loss. A schematic is shown in Figure 1.

This procedure has two key decisions: 1) where to logically split the teacher and student models, and 2) the choice of IR loss. We discuss these settings in the hyperparameter optimization below.

**Hyperparameter optimization.** LIT inherits two hyperparameters from KD and introduces one more: $\tau$ (the temperature in KD), $\alpha$ (the interpolation parameter in KD), and $\beta$ (the interpolation parameter in LIT), along with an intermediate representation loss and split points. In this work, we only consider adding the IR loss between natural split points, e.g., when a downsampling occurs in a convolutional network. We have additionally found that L2 loss works well in practice, so we use the L2 loss for all experiments unless otherwise noted (Section 4.4).

We have found that iteratively setting $\tau$, then $\alpha$, then $\beta$ to work well in practice. We have found that the same hyperparameters work well for a given student and teacher structure (e.g., ResNet teacher and ResNet student). Thus, we use the same set of hyperparameters for a given student and teacher structure (e.g., we use the same hyperparameters for a teacher/student of ResNet-110/ResNet-20 and ResNet-110/ResNet-32). To set the hyperparameters for a given structure, we first set $\tau$ using a small student model, then $\alpha$ for the fixed $\tau$, then $\beta$ for the fixed $\alpha$ and $\tau$ (all on the validation set).

## 4  EXPERIMENTS

**Experimental setup.** We evaluate LIT's efficacy at compressing models on a range of tasks and models, including image classification, sentiment analysis, and image-to-image translation (GAN). Throughout, we use student and teacher networks with the same broad architecture (e.g., ResNet to ResNet). We consider ResNet (He et al., 2016), ResNeXt (Xie et al., 2017), VDCNN (Conneau et al., 2016), and StarGAN (Choi et al., 2017). We use standard architecture depths, widths, and learning rate schedules (described in the Appendix), and perform hyperparameter selection for the KD and LIT interpolation parameters, while also performing a sensitivity analysis of these hyperparameters in the sequel.

**Comparison to pruning.** In this work we focus on *modern* networks, which pruning does not perform well on. For example, Han et al. (2015a) compresses VGG, where the majority of the weights are in the fully connected (FC) layer, achieving a $10\times$ compression rate for the FC layer, but only a $1.14\times$ compression rate for the convolutional layers. In contract, ResNet-110 can only be pruned by ~$1.6\times$ (Li et al., 2016), compared to LIT's $4\times$ compression for ResNet-110.

### 4.1  LIT SIGNIFICANTLY COMPRESSES MODELS

**LIT is effective at compressing a range of datasets and models.** We ran LIT on a variety of models and datasets for image classification and sentiment analysis (Table 1). We additionally performed KD and hint training on these datasets and models. We selected the hyperparameters sequentially (Section 3).

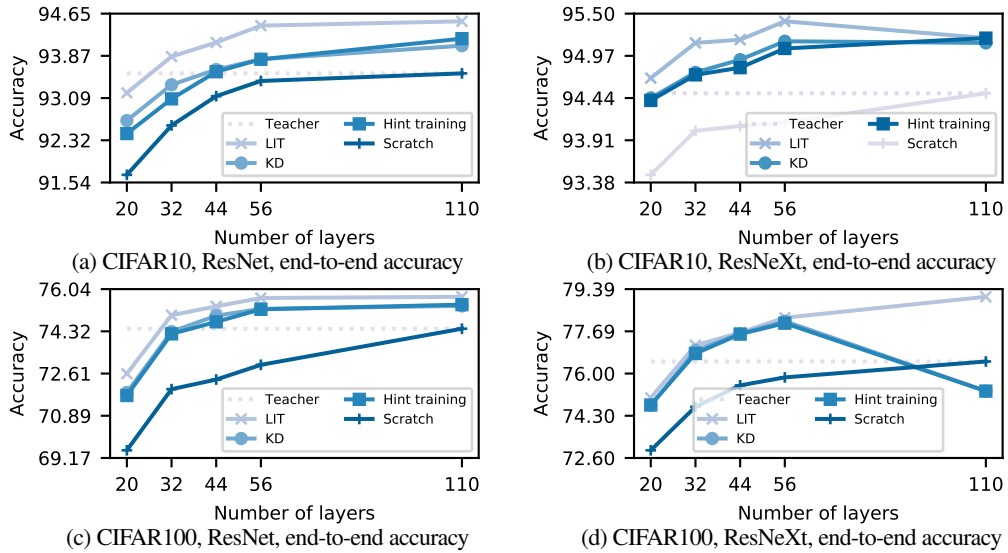

Figure 2: The accuracy of ResNet and ResNeXt trained from scratch, trained via KD, and trained via LIT for CIFAR10/100. The teacher model was ResNet-110 and ResNeXt-110 respectively. As shown, LIT outperforms KD for every student model. Identical student and teacher architectures corresponds to born again networks, which LIT also outperforms. In some cases, KD can reduce the accuracy of the student model, as reported in Mishra & Marr (2017).

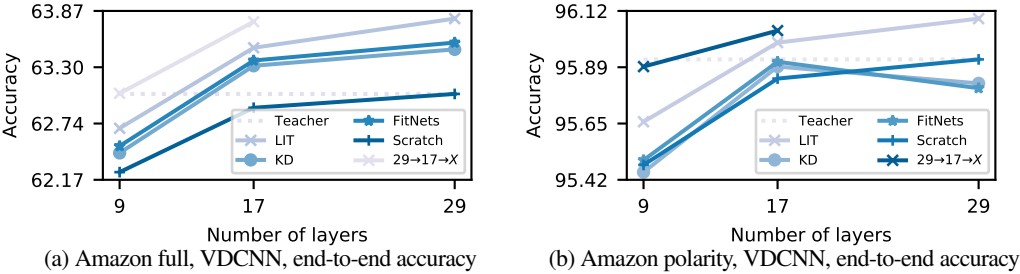

Figure 3: The accuracy of VDCNN on Amazon reviews (full and polarity) trained from scratch, trained via KD, and trained via LIT.

Figure 2 shows the results for ResNet and ResNeXt for CIFAR10 and CIFAR100, and Figure 3 shows the results for VDCNN on Amazon Reviews (full, polarity). LIT can compress models by up to $5.5\times$ (CIFAR10, ResNeXt 110 to 20) on image classification and up to $3.2\times$ on sentiment analysis (Amazon Reviews, VDCNN 29 to 9), with no loss in accuracy. LIT outperforms KD and hint training on all settings. Additionally, LIT outperforms the recently proposed Born Again procedure in which the same architecture is used as both the student and teacher model (Furlanello et al., 2018) (i.e., only for improved accuracy, not for compression). We have found that, in some cases, training sequences of models using LIT results in higher performance. Thus, for VDCNN, we additionally compressed using LIT a VDCNN-29 to a VDCNN-17, and using this VDCNN-17, we trained a VDCNN-9 and VDCNN-17.

We also found that in some cases, KD degrades the accuracy of student models when the teacher model is the same architecture (ResNeXt-110 on CIFAR100, VDCNN-29 on Amazon Reviews polarity). This corroborates prior observations in Mishra & Marr (2017).

**LIT can reduce group cardinality.** While LIT requires the size of at least one IR to be the same width between the teacher and student model, several classes of models have an internal width or group *cardinality*. For example, ResNeXt (Xie et al., 2017) has a "grouped convolution", which is equivalent to several convolutions with the same input (see Figure 3 in Xie et al. (2017)). The width of the network is not affected by the group size, so LIT is oblivious to the group size.

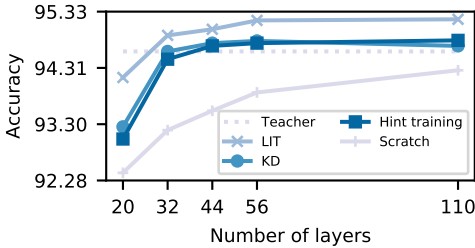

Figure 4: ResNeXt student models with cardinality 16 trained from a ResNeXt-110 with cardinality 32 on CIFAR10. We show that LIT can reduce the cardinality and that LIT outperforms KD.

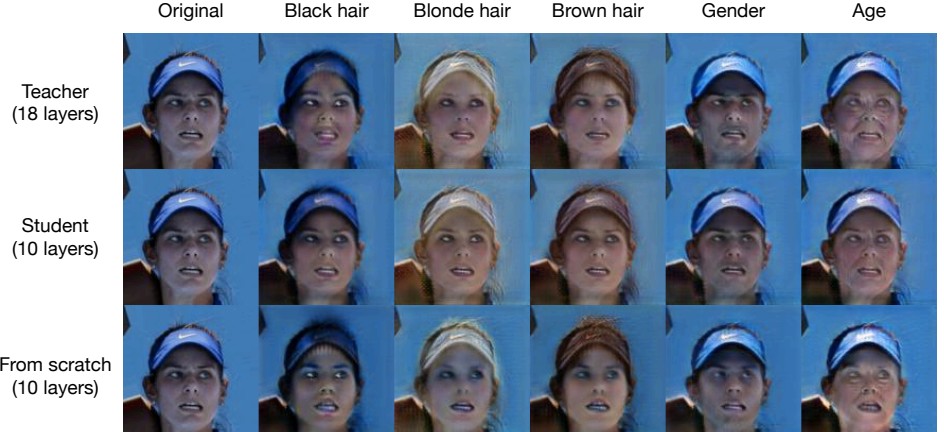

Figure 5: Selected images from the teacher (six residual blocks), student (two residual blocks), and trained from scratch (two residual blocks) StarGANs. As shown (column two, four), LIT can appear to improve GAN performance while significantly compressing models. We show a randomly selected set of images in the Appendix. Best viewed in color.

| Model | Inception score (higher is better) | FID score (lower is better) |
|---|---|---|
| Teacher (18 layers) | 3.49 | 6.43 |
| LIT student (10 layers) | **3.56** | **5.84** |
| L2 student (10 layers) | 3.46 | 6.47 |
| Trained from scratch (10 layers) | 3.37 | 6.56 |
| Randomly initialized (10 layers) | 2.63 | 94.00 |
| Randomly initialized (18 layers) | 2.45 | 151.43 |

Table 2: Inception Salimans et al. (2016) and FID Heusel et al. (2017) scores for different versions of StarGAN. Despite having fewer layers than the teacher, the LIT student model achieves the best scores.

We show that LIT can reduce the group cardinality for ResNeXt. We train student ResNeXts with cardinality 16 (instead of the default 32) from a ResNeXt-110 (cardinality 32) on CIFAR10. Figure 4 illustrates the results. As before, LIT outperforms KD and hint training in this setting.

**LIT can compress GANs.** We compressed StarGAN's generator (Choi et al., 2017) using the LIT procedure with $\beta = 0$ (i.e., only using the intermediate representation loss) and with an analogous procedure to KD with the L2 loss as a baseline. The original StarGAN has 18 total convolutional layers (including transposed convolutional layers), with 12 of the layers in the residual blocks (for a total of six residual blocks). We compressed the six residual blocks to two residual blocks (i.e., 12 to four layers) while keeping the rest of the layers fixed. The remaining layers for the teacher model were copied to the student model and fine-tuned. The discriminator remained fixed.

As shown in Table 2, LIT outperforms all baselines in inception Salimans et al. (2016) and FID Heusel et al. (2017) scores. Additionally, as shown in Figure 5, the student model appears to perceptually outperform both the teacher model and equivalent model trained from scratch, suggesting LIT can both compress GANs and serve as a form of regularization.

| Type | Accuracy | Type | Accuracy |
|---|---|---|---|
| LIT | **93.25%** | LIT | **94.72%** |
| KD | 92.75% | KD | 94.42% |
| One IR, teacher input | 92.74% | One IR, teacher input | 94.21% |
| One IR, no teacher input (FitNets) | 92.68% | One IR, no teacher input (FitNets) | 94.18% |
| Multiple IRs, no teacher input | 90.42% | Multiple IRs, no teacher input | 91.27% |

Table 3: Ablation study of LIT. We performed LIT, KD, and three modifications of LIT. The second block was used for single IR experiments. As shown, LIT outperforms KD and the modifications, while all the modifications underperform standard KD. **Left:** ResNet, **Right:** ResNeXt.

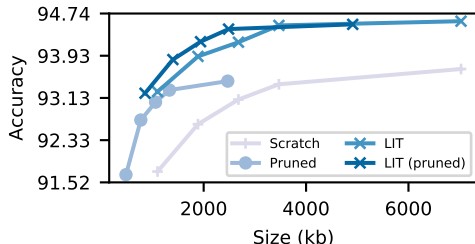

Figure 6: The size vs accuracy of various ResNets pruned on CIFAR10. LIT outperforms standard pruning Han et al. (2015b).

| Model | Loss | Accuracy | Model | Loss | Accuracy |
|---|---|---|---|---|---|
| ResNet | L2 | 93.20±0.04 | ResNeXt | L2 | 94.63±0.07 |
| ResNet | L1 | 93.19±0.05 | ResNeXt | L1 | 94.62±0.07 |
| ResNet | Smoothed L1 | 93.02±0.06 | ResNeXt | Smoothed L1 | 93.86±0.08 |

Table 4: Effect of intermediate representation loss on student model accuracy. L2 and L1 do not significantly differ, but smoothed L1 degrades accuracy. Average of three runs on CIFAR10.

## 4.2 IMPACT OF TRAINING TECHNIQUES

LIT uses block-wise training with the teacher IRs as input to the student model. To show the effectiveness of block-wise training, we tried other variations: 1) matching a single IR, with no input from the teacher (i.e., standard hint training/FitNets), 2) a single IR with teacher input, 3) multiple IRs with no teacher input. We performed these variations on a teacher model of ResNet-110 and a student model of ResNet-20 on CIFAR10 and similarly for ResNeXt. We used the second block for the single IR experiments.

As shown in Table 3, none of the three variants are as effective as LIT or KD. Thus, we see that LIT's block-wise training is critical for high accuracy compression.

## 4.3 LIT IS COMPLEMENTARY TO PRUNING

Pruning is a key technique in deep compression in which parts of a network are set to zero, which reduces the number of weights and, on specialized hardware, reduces the computational footprint of networks. To see if LIT models are amenable to pruning, we pruned ResNets trained via LIT. We additionally pruned ResNets trained from scratch. All experiments were done on CIFAR10 using a standard pruning procedure Han et al. (2015b).

As shown in Figure 6, LIT models outperform standard pruning for accuracy at a given model size. Additionally, LIT models can be pruned, although less than their trained-from-scratch counterparts. However, LIT models are more accurate and are thus likely learning more meaningful representations. Thus, we expect LIT models to be more difficult to prune, as each weight is more important.

## 4.4 SENSITIVITY ANALYSIS OF HYPERPARAMETERS

**Intermediate loss penalty.** To see the affect of the intermediate loss penalty, we performed LIT from a teacher model of ResNet-110 to a student of ResNet-20 with the L1, L2, and smoothed L1 loss (all on CIFAR10). The results are shown in Table 4. As shown, L2 and L1 do not significantly differ ($p=0.78$), but smoothed L1 degrades accuracy ($p=0.02$).

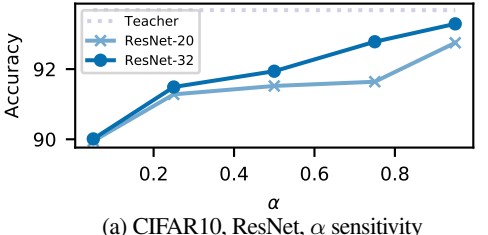 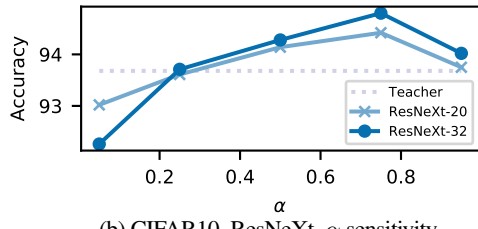

(a) CIFAR10, ResNet, $\alpha$ sensitivity        (b) CIFAR10, ResNeXt, $\alpha$ sensitivity

Figure 7: The accuracy of student models as $\alpha$ (KD's interpolation factor for the cross-entropy and logit loss) varies for ResNet and ResNeXt on CIFAR10. The optimal $\alpha$ varies by model type.

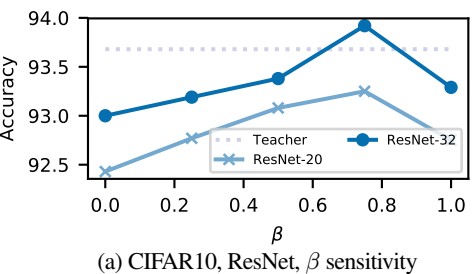 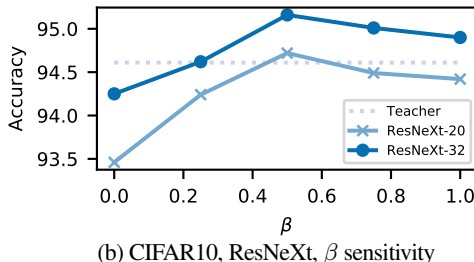

(a) CIFAR10, ResNet, $\beta$ sensitivity        (b) CIFAR10, ResNeXt, $\beta$ sensitivity

Figure 8: The accuracy of student models as $\beta$ (LIT's interpolation factor between KD loss and IR loss) varies for ResNet and ResNeXt on CIFAR10. As shown, LIT outperforms training only via KD ($\beta{=}1$) and only via intermediate representations ($\beta{=}0$). The optimal $\beta$ appears to be lower (i.e., closer to only using the intermediate representation loss) for more accurate models; we hypothesize that more accurate models learn more informative intermediate representations, which helps the students learn better.

| Model | Precision | Accuracy | Model | Precision | Accuracy |
|-------|-----------|----------|-------|-----------|----------|
| ResNet | fp32 | 93.20±0.04 | ResNeXt | fp32 | 94.63±0.07 |
| ResNet | Mixed | 93.17±0.07 | ResNeXt | Mixed | 94.57±0.10 |

Table 5: Affect of mixed-precision training on the LIT procedure. Mixed-precision training does not significantly affect the accuracy of the LIT procedure. Average of three runs on CIFAR10.

$\alpha$ **and** $\beta$**.** Recall that $\alpha$ is the weighting parameter in KD and $\beta$ is the relative weight of KD vs the intermediate representation loss (Section 3).

To see the effect of of $\alpha$ (which is a KD hyperparameter), we varied $\alpha$ between 0 and 1 for ResNet and ResNeXt on CIFAR10 and CIFAR100. As shown in Figure 7, $\alpha$ can significantly affect accuracy. Thus, we searched for $\alpha$ as opposed to using a static policy of 0.5 as in Hinton et al. (2014).

We additionally varied $\beta$ between 0 and 1 for ResNet and ResNeXt on CIFAR10. As shown in Figure 8, the optimal $\beta$ varies between architectures but appears to be consistent within the same meta-architecture.

**LIT works with mixed precision.** To confirm mixed precision training (Micikevicius et al., 2017) works with LIT, we ran LIT on ResNet and ResNeXt (the teacher had 110 layers and the student had 20 layers) on CIFAR10 with both fp32 and mixed precision training. The results are shown in Table 5. As shown, mixed precision does not significantly change the results for ResNet or ResNeXt ($p{=}0.5, 1.0$ respectively).

## 5 CONCLUSION

We introduce LIT, a novel model compression technique that trains a student model from a teacher model's intermediate representations. LIT requires at least one intermediate layer of the student and teacher to match in width, which allows parts of the teacher model to be copied to the student model. By combining several such intermediate layers, LIT students learn a high quality representation of the teacher state without the associated depth. To overcome the lack of useful intermediate representations within the student model at the beginning of training, LIT uses the teacher's intermediate representations as input to the student model during training. We show that LIT can compress models up to $5.5\times$ with no loss in accuracy on standard classification benchmark tasks and image-to-image translation (i.e., GAN generators), outperforming standard KD and hint training.

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

| Model | Parameterization |
|---|---|
| ResNe(X)t-20 | [3, 3, 3] |
| ResNe(X)t-32 | [5, 5, 5] |
| ResNe(X)t-44 | [7, 7, 7] |
| ResNe(X)t-56 | [9, 9, 9] |
| ResNe(X)t-110 | [18, 18, 18] |

Table 6: ResNet and ResNeXt types for CIFAR10/100.

## A  EXPERIMENTAL DETAILS

All network architectures we use are standard architectures for the datasets of choice. All student/teacher pairs were of the same architecture type (e.g., ResNet to ResNet). For each dataset, we detail the architecture and hyperparameters used.

All experiments were performed in PyTorch v0.4.0 with Python 3.5. Experiments were run on public cloud and custom servers using NVIDIA P100, V100, Titan Xp, and Titan V GPUs.

All weights for architectures were initialized as in the original architecture.

### A.1  CIFAR10/100

We used the same hyperparameters for CIFAR10 and CIFAR100.

**ResNet.** We use standard ResNets (He et al., 2016) for CIFAR10/100 (Krizhevsky & Hinton, 2009; **?**). The architectures are parameterized by the number of "computation heavy" layers, i.e., convolutional and fully connected layers, but not batch norm and ReLU layers. Thus, a ResNet-110 has 110 convolutional and fully connected layers.

Each ResNet has three blocks, where the convolutional layers in each block have the same number of filters. The last convolutional layer in each block downsamples by a factor of two. The ResNets can be parameterized by the number of residual blocks in each block, where each residual block has two convolutional layers. This parameterization is the same parameterizing by the number of layers. For example, an [18, 18, 18] has 110 layers total, 108 layers in the blocks, along with an additional convolutional layer at the start and a fully connected layer. We show a table of the number of layers and block parameterizations in Table 6.

For all experiments, we used a batch size of 32, SGD with a momentum of 0.9, and weight decay of 1e-4.

For training from scratch, we trained with a starting learning rate of 0.1 for 200 epochs with milestones at 100 and 150 epochs, decaying the learning rate by a factor of 10.

For KD, we trained with a starting learning rate of 0.1 for 250 epochs with milestones at 100 and 175 epochs, decaying the learning rate by a factor of 10. We used $\tau = 6$ and $\alpha = 0.95$.

For LIT, we trained with a starting learning rate of 0.1 for 175 epochs with milestones at 60, 100, and 125 epochs, decaying the learning rate by a factor of 10. We then fine-tuned using the KD loss for another 75 epochs with a starting learning rate of 0.01 and milestones at 35 and 55 epochs. We used $\beta = 0.75$.

**ResNeXt.** We use standard ResNeXts (Xie et al., 2017) for CIFAR10/100. The parameterization in terms of number of layers are the same for ResNet.

ResNeXt has an additional parameter of the group cardinality. We use a group cardinality of 32 for all experiments, except the ones detailed below.

For all experiments, we used a batch size of 32, SGD with a momentum of 0.9, and weight decay of 1e-4.

For training from scratch, we trained with a starting learning rate of 0.1 for 300 epochs with milestones at 150 and 225 epochs, decaying the learning rate by a factor of 10.

For KD, we trained with a starting learning rate of 0.1 for 300 epochs with milestones at 100, 175, and 225 epochs, decaying the learning rate by a factor of 10. We used $\tau = 6$ and $\alpha = 0.95$.

For LIT, we trained with a starting learning rate of 0.1 for 200 epochs with milestones at 100, 145, and 175 epochs, decaying the learning rate by a factor of 10. We then fine-tuned using the KD loss for another 125 epochs with a starting learning rate of 0.01 and milestones at 65, 95, and 110 epochs. We used $\beta = 0.5$.

**Reduced cardinality ResNeXt.**

All hyperparameters were the same as the standard ResNeXt experiments except we used $\beta = 0.25$ and a student cardinality of 16.

## A.2 AMAZON REVIEWS

We use standard VDCNNs (Conneau et al., 2016) for Amazon Reviews full and polarity (He & McAuley, 2016). VDCNN has an initial convolutional layer and four blocks of convolutional layers (each block has the same number, but vary between types of VDCNNs). Thus, a VDCNN-9 has an initial convolutional layer and two convolutional layers in each subsequent block. We consider VDCNN-9, VDCNN-17, and VDCNN-29 as in Conneau et al. (2016).

For all experiments, we used a batch size of 128, SGD with a momentum of 0.9, and weight decay of 1e-4.

For training from scratch, we trained with a starting learning rate of 0.01 for 15 epochs, with milestones at 3, 6, 9, 12, and 15 epochs, decaying the learning rate by a factor of 10.

For KD, we trained with a starting learning rate of 0.01 for 18 epochs, with milestones at 3, 6, 9, 12, and 15 epochs, decaying the learning rate by a factor of 10. We used $\tau = 6$ and $\alpha = 0.98$.

For LIT, we trained with a starting learning rate of 0.01 for 15 epochs, with milestones at 3, 6, 9, and 12 epochs, decaying the learning rate by a factor of 2. We then fine-tuned using the KD loss for another 10 epochs with a starting learning rate of 0.000625 and milestones at 4 and 8 epochs. We used $\beta = 0.02$.

## A.3 STARGAN

We used the StarGAN as described in Choi et al. (2017). The original StarGAN has 18 total convolutional layers (including transposed convolutional layers), with 12 of the layers in the residual blocks (each residual block has two convolutional layers). We compressed the six residual blocks to two residual blocks.

For all experiments, we used a batch size of 16, SGD with a momentum of 0.9, and weight decay of 1e-4.

For training from scratch, we trained with a starting learning rate of 0.0001 for 20 epochs. The learning rate was decayed to 0 over the last 10 epochs.

For LIT, we trained with a starting learning rate of 0.0001 for 16 epochs, decaying the learning rate by 10 at epoch 8 (only the IR loss was used). We then fine-tuned with the discriminator with a starting learning rate of 0.00005 for 10 epochs, decaying the learning rate by 10 at epoch 5.

## A.4 STATISTICAL TEST DETAILS

p-values were computed with a t-test Wasserman (2013) with three samples each.

All error estimates are the standard deviation with three samples.

## B STARGAN IMAGES

We show a randomly selected set of images generated from the StarGAN teacher, student, and trained from scratch generators in Figure 9.

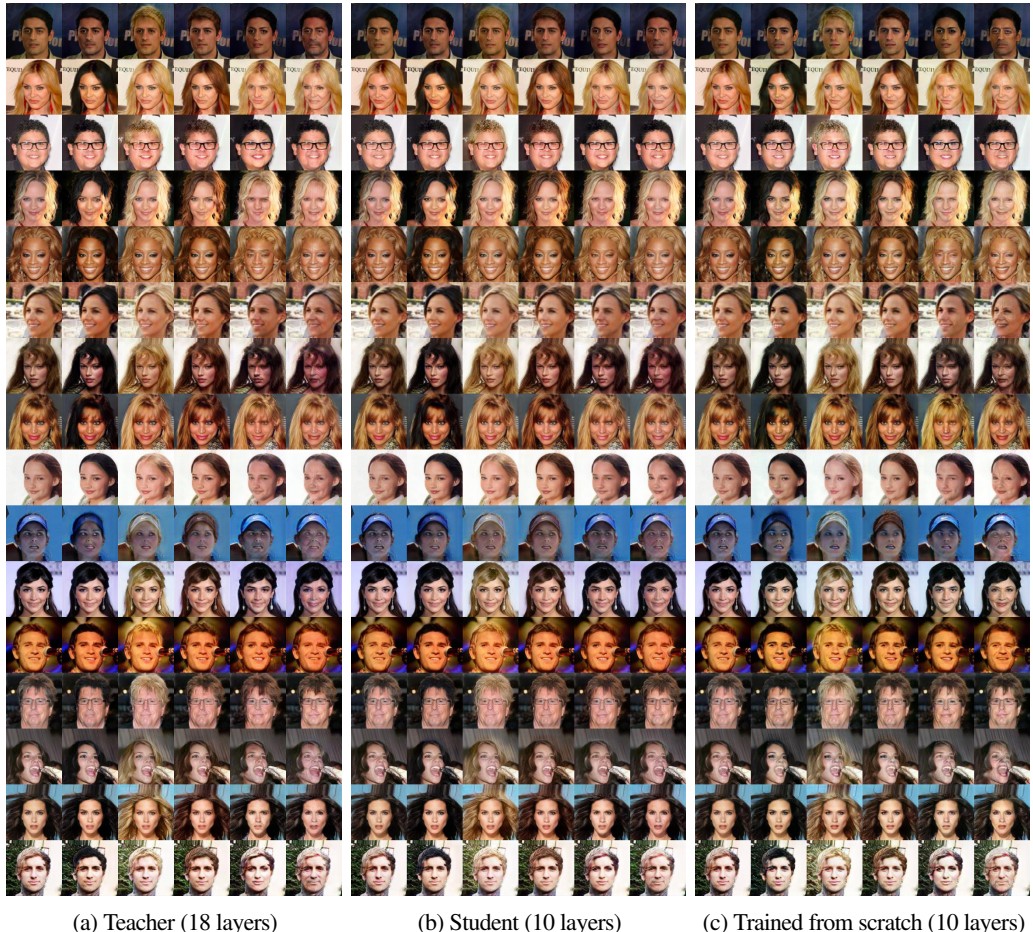

(a) Teacher (18 layers)          (b) Student (10 layers)          (c) Trained from scratch (10 layers)

Figure 9: Randomly selected images from the teacher (six residual blocks, 18 total layers), student (two residual blocks, 10 total layers), and trained from scratch (two residual blocks, 10 total layers) StarGANs. As shown, LIT can appear to improve GAN performance while significantly compressing models. The columns are: Original, Black Hair, Blond Hair, Brown Hair, Male, Age.

