# OpenReview forum: "LIT: Block-wise Intermediate Representation Training for Model Compression"
_ICLR.cc/2019/Conference_

### Official Review · AnonReviewer1 · 2018-11-02
**paper well presented, experimental validation could be further improved**

**Rating:** 6
**Confidence:** 4

**Review:**

This paper introduces LIT, a network compression framework, which uses multiple intermediate representations from a teacher network to guide the training of a student network. Experiments are designed such that student networks are shallower than teacher networks, while maintaining their width. The method is validated on CIFAR-10 and 100 as well as on Amazon Reviews.

The paper is clearly written and easy to follow. The main novelty of the paper is essentially using the teacher intermediate representations as input to the student network to stabilize the training, and applying the strategy to recent networks and tasks.

The authors claim that they are only concerned with knowledge transfer between layers of the same width, that is teacher and student network been designed (by model construction) to have the same number of downsampling operations, while maintaining the same number of stages (referred to as sections in the paper). However, resnet-based architectures have been shown to perform iterative refinement of their features between downsampling operations (see e.g. https://arxiv.org/pdf/1612.07771.pdf and https://arxiv.org/pdf/1710.04773.pdf ). Moreover, these models were also shown to be good regularizers, since they can reduce their model capacity as needed (see https://arxiv.org/pdf/1804.11332.pdf).  Therefore, having experiments skipping stages would be interesting, and may allow to further compress the networks (by skipping layers or stages which do not incorporate much transformation). Following https://arxiv.org/pdf/1804.11332.pdf, for the sake of completeness, it might also be interesting to compare LIT results to the ones obtained by just removing layers in the teacher network which have small weight norms.

In method, the last sentence before "knowledge distillation loss" suggests the training of student networks might not be done end-to-end. Could the authors clarify this?
It seems there might be a typo in the KD loss of "knowledge distillation loss", equation (2). Shouldn't the second term of the equation be a function of p^T and q^T (with temperature)?

I would suggest changing "sections" to stages, as previously introduced in https://arxiv.org/pdf/1612.07771.pdf .

As for the experiments, it would be more interesting to see this kind of analysis on ImageNet (pretained resnet models are readily available).
Figure 3, why not add hint training as well?
Figure 4, what's the dataset used here?

In Section 4.2, it seems that the choice of the IR layer in the analysis could have a significant impact. How was the layer chosen for the ablation study experiments?

There are a few overstatements in the paper:
- page 5, paragraph 2: FitNets proposes a general framework to transfer knowledge from a teacher network to a student network through intermediate layers. Thus, the framework itself does not require the student networks to be deeper and thinner than the teacher network.
- page 6, "LIT can compress GANs": authors claim to overcome limitations of KD when it comes to applying knowledge transfer to pixel-wise architecture that do not output distributions. It seems that changing the loss and using a l2 loss instead is a rather minor change, especially since performing knowledge transfer by means of l2 (although at intermediate layers) has already been explored in FitNets.

Please add references for inception and FID scores.
Please fix references format in page 10.

---

> ### Author Response · Authors · 2018-11-08
> **Thank you for your review**
>
> Thank you for the thoughtful review. We have responded to your comments inline. We have improved the manuscript based on your feedback.
>
> 1. Compare LIT to removing small weight norm parts of networks.
>
> Li et al. 2017 (https://arxiv.org/pdf/1608.08710.pdf) removes small norm filters from networks, including ResNets. They achieve ~1.6x compression for the same ResNets we use in this paper, which significantly underperforms LIT. We have added this reference to the paper.
>
>
> 2. Clarifying the training procedure.
>
> LIT trains with the combined loss for some number of epochs. Then, LIT trains with just the KD loss after that. We have clarified this in the manuscript.
>
>
> 3. Typo in Eq. 2.
>
> We have fixed the typo.
>
>
> 4. Changing sections to stages.
>
> Thank you for pointing out the standard terminology. We have updated sections to stages in the manuscript.
>
>
> 5. ImageNet models
>
> Unfortunately training ImageNet models and hyperparameter tuning alpha and beta are computationally expensive. We are currently running these experiments, but they may not complete by the revision close period.
>
>
> 6. Hint training in Figure 3.
>
> We were unable to complete hint training experiments in time for the submission, but they have completed. We have added hint training to Figure 3. Briefly, hint training outperforms KD, but underperforms LIT.
>
>
> 7. Dataset in Figure 4.
>
> For Figure 4, we used CIFAR10. We have updated the caption to reflect this.
>
>
> 8. Choice of IR.
>
> We used the IR after the second stage of the ResNet. We have updated the manuscript to reflect this.
>
>
> 9. Overstatements in the paper.
>
> We have fixed these statements in the paper.
>
> We realized the issue for GANs in the paper and conducted the L2 experiment (i.e., KD with a different loss). As we show in the updated paper (Table 2), LIT outperform this procedure.
>
>
> 10. References and formatting.
>
> We have added references to the Inception and FID scores. We have additionally fixed the formatting on the last page of citations.

---

### Official Review · AnonReviewer2 · 2018-11-03
**A novel approach for compressing deep learning models**

**Rating:** 6
**Confidence:** 3

**Review:**

This paper proposes to compress the model by depth. It uses hint training and knowledge distillation techniques to compress a "deep" network block-wisely. It shows a better compression ratio than knowledge distillation or hint training while achieving comparable accuracy performance.

Pros:
1. This paper considers block-wise compression. For each block, it uses the output of the teacher's last layer as input during training, which improves the learnability of the student models.
2. The experiments include a large range of tasks, e.g., image classification, sentiment analysis and GAN.

Cons:
1. Validation accuracy is used as the performance metric, which might be over-tuned. How is the performance on testing datasets?
2. The writing and organization of the paper need some improvement, especially the experiments section.
3. The compression ratio (3-5) is not very impressive compared with other compression techniques with pruning and quantization techniques, such as Han et al. 2015, Hubara et al. 2016.

In summary, I think this is an interesting approach to compress deep learning models. But I think the comparisons should be done in terms of testing accuracy. Otherwise, it is hard to judge the performance of this approach.

=== after rebuttal ===
Thanks for the authors' response. Some of my concerns have been clarified. I increased my rating from 5 to 6.

---

> ### Author Response · Authors · 2018-11-08
> **Thank you for the review; initial response**
>
> Thank you for the thoughtful review. We have responded to your comments inline. We have improved the manuscript based on your feedback. Our experiments using a test dataset are in progress and we will update the manuscript upon completion.
>
> 1. Validation accuracy is used as the performance metric, which might be over-tuned. How is the performance on testing datasets?
>
> Thank you for your thoughtful question. We agree with your point that validation accuracy may be over-tuned. We have started to run experiments with a separate test set, which will take some time due to our limited computational resources. We have initial results for ResNet on CIFAR10, which also show that LIT outperforms training from scratch, and KD. The results are essentially the same as the results currently in the manuscript. For ResNet-110 -> ResNet-20 we found that:
> - LIT achieves 93.19%,
> - KD achieves 92.68%,
> - Training from scratch achieves 91.68%
> Once we have completed the rest of the results, we will update the manuscript with test accuracy.
>
> We note that the majority of compression papers (including Han et al. 2015 and Hubera et al. 2016, Li et al. 2017 mentioned below, Furlanello et al. 2018, etc.) and the original ResNet and ResNeXt papers use validation accuracy as their primary metric. Additionally, Li et al. 2017 and Conneau et al. 2017 (the original VDCNN paper) refer to validation accuracy as “test accuracy.” To ensure LIT can be compared against other methods, we will also report validation accuracy, as using a separate test set requires using a different set of data.
>
>
> 2. The writing and organization of the paper need some improvement, especially the experiments section.
>
> We have improved the presentation of the experiments section by removing some redundancy, pointing to the appendix for further experimental details, and adding details for which datasets were used. Are there other points we should address?
>
>
> 3. The compression ratio (3-5) is not very impressive compared with other compression techniques with pruning and quantization techniques, such as Han et al. 2015, Hubara et al. 2016.
>
> Both Han et al. 2015 and Hubara et al. 2016 test on older networks (e.g., VGG) where the majority of the weights are in the fully connected layers. Compressing the FC layer can achieve up to ~10x compression, while compressing the convolutional layers achieves around ~1.14x compression. As the majority of weights for these older networks are in the FC layer, this achieves high compression rates for these networks.
>
> Compressing modern networks is significantly harder. For example, Li et al. 2017 (https://arxiv.org/pdf/1608.08710.pdf) only achieves ~1.6x compression on ResNet (which achieves significantly higher accuracy than VGG). We believe our results should be compared against other methods for compressing _modern_ networks. We have made this point more clear in the paper.
>
> Additionally, in this work, we focus on compression techniques that can improve inference throughput on existing hardware. Pruning and quantization generally require special hardware (e.g., Han et al. 2016’s EIE https://arxiv.org/abs/1602.01528) for inference improvements.

---

### Official Review · AnonReviewer3 · 2018-11-08
**cute idea but need more analysis**

**Rating:** 5
**Confidence:** 4

**Review:**

This paper proposes a new approach to compress neural networks by training the student's intermediate representation to match the teacher's.

The paper is easy to follow. The idea is simple. The motivation and contribution are clear. The experiments are comprehensive.

One advantage of the proposed approach that the authors did not mention is that LIT without KD can be optimized in parallel, though I'm not sure how useful this is.

One major weakness of the paper is how the hyperparameters, such as the number of layers, the alpha, beta, tau, and so on, are tuned. It is not clear from the paper that there is a separate development set for tuning these values. If the hyperparameters are tuned on the test set, then it is not surprising LIT works better.

Here are some minor questions:

p.5

LIT outperforms KD and hint training on all settings.
--> what are the training errors (cross entropy) for LIT, KD and hint training? what about the KD objectives (on the training set) of the model trained with LIT and the one trained with KD? this might tell us why LIT is better than the two.

LIT outperforms the recently proposed Born Again procedure ...
--> what are the training errors (cross entropy) before and after the born again procedure? this might help us understand why LIT is better.

KD degrades the accuracy of student models when the teacher model is the same architecture
--> again, the training errors (cross entropy) might be able to help us understand what is going on.

p.7

As shown in Table 3, none of the three variants are as effective as LIT or KD.
--> is this claim statistically significant? some of the differences are very small.

We additionally pruned ResNets trained from scratch.
--> what pruning method is being used?

As shown in Figure 6., LIT models are pareto optimal in accuracy vs model size.
--> this is a very strong claim. it's better to say we fail to prune the network with the approach, but we don't know whether there exists another approach that can reduce the network size while maintaining accuracy.

As shown, L2 and L1 do not significantly differ, but smoothed L1 degrades accuracy.
--> is this claim statistically significant?

---

> ### Author Response · Authors · 2018-11-08
> **Thank you for your review; initial response**
>
> Thank you for the thoughtful review. We have responded to your comments inline. We have improved the manuscript based on your feedback. Several experiments are in progress and we will update the manuscript upon completion.
>
> 1. Hyperparameter tuning
>
> The hyperparameters of alpha and tau are directly taken from KD. The only hyperparameter LIT introduces is beta. We are in the process of updating our results when using separate validation set for hyperparameter selection and test set (see the response to reviewer number 2). Our initial results show that LIT outperforms KD and training from scratch by the same margins.
>
>
> 2. Further analysis of training errors
>
> Thank you for the suggestion. We are in the process of conducting this analysis and will respond once we have completed this analysis.
>
>
> 3. Differences in table 3 are small
>
> We are in the process of running the training procedure multiple times and will perform a statistical test upon completion. We will update the manuscript once the analysis has completed. However, the trend of LIT outperforming KD is consistent across architectures (ResNet, ResNeXt, VDCNN), datasets (CIFAR10, CIFAR100, Amazon Reviews), and tasks (image classification, sentiment analysis). Additionally, a 0.5% increase in accuracy corresponds to nearly doubling the depth of the network and corresponds to a 7% reduction in error.
>
>
> 4. Pruning method for LIT.
>
> We used standard pruning proposed by Han et al. 2015 (https://arxiv.org/abs/1506.02626), in which small weights are iteratively removed and the network is retrained at each step. We have updated the manuscript to reflect this.
>
>
> 5. LIT vs pruning
>
> Thank you for the comment. We have updated the manuscript to avoid overclaiming. Additionally, pruning typically requires new hardware for improved inference throughput, whereas LIT does not.
>
>
> 6. Statistical significance of different loss functions.
>
> We are in the process of running the training procedure multiple times and will perform a statistical test upon completion. Once we have the results, we will update the manuscript.

---

> > ### Author Response · Authors · 2018-11-24
> > **Statistical significance**
> >
> > We have conducted a statistical test for significance for the differences throughout the paper (except table 3, see below). Our conclusions are now supported with p-values in the updated manuscript.
> >
> > All the differences that we have measured in table 3 are significant. We are in the process of running multiple trials of KD and will update the manuscript when it has finished.

---

> > > ### Comment · AnonReviewer3 · 2018-12-04
> > > **score revision**
> > >
> > > Thanks for the update. The revised paper reads much better than the original submission. I wish there could be more analyses as to why/how LIT works. I have revised my score accordingly.

---

### Author Response · Authors · 2018-11-24
**Further results with test accuracy**

Dear reviewers,

We have updated our manuscript with test accuracy for CIFAR10 (Figure 2a, 2b). We are in the process of running the other experiments for test accuracy and they will be complete by the camera-ready due date. As shown, the results do not significantly differ when we hyperparameter tune on a validation set and test on a separate test set.

Best,
LIT team

---

> ### Author Response · Authors · 2018-11-27
> **CIFAR100 test accuracy results**
>
> We have updated our manuscript with test accuracy results for CIFAR100 (Figure 2c, 2d). As shown, the results have not significantly changed, and LIT outperforms all baselines.
>
> Best,
> LIT team

---

### Meta-Review · Area_Chair1 · 2018-12-14
**Modifies knowledge distillation by training student to match teachers intermediate representation at multiple layers.**

**Confidence:** 4
**Recommendation:** Reject

**Metareview:**

The authors propose a method for distilling a student network from a teacher network and while additionally constraining the intermediate representations from the student to match those of the teacher, where the student has the same width, but less depth than the teacher. The main novelty of the work is to use the intermediate representation from the teacher as an input to the student network, and the experimental comparison of the approach against previous work.

 The reviewers noted that the method is simple to implement, and the paper is clearly written and easy to follow. The reviewers raised some concerns, most notably that the authors were using validation accuracy to measure performance, and were thus potentially overfitting to the test data, and regarding the novelty of the work. Some of the criticisms were subsequently amended in the revised version where results were reported on a test set (the conclusions are as before).  Overall, the scores for this paper were close to the threshold for acceptance, and while it was a tough decision, the AC ultimately felt that the overall novelty of the work was slightly below the acceptance bar.